# An Overview of the Treatment Strategy of Esophagogastric Junction Cancer

**DOI:** 10.3390/cancers17121961

**Published:** 2025-06-12

**Authors:** Masatoshi Nakagawa, Masanobu Nakajima, Masaki Yoshimatsu, Yu Ueta, Noboru Inoue, Takahiro Ochiai, Shuhei Takise, Junki Fujita, Shinji Morita, Kazuyuki Kojima

**Affiliations:** Department of Upper Gastrointestinal Surgery, Dokkyo Medical University, 880 Kitakobayashi, Mibu-machi, Shimotsuga-gun, Tochigi 321-0293, Japan; mnakajim@dokkyomed.ac.jp (M.N.); my-1157@dokkyomed.ac.jp (M.Y.); u-yuu19@dokkyomed.ac.jp (Y.U.); ninoue@dokkyomed.ac.jp (N.I.); t-ochiai213@dokkyomed.ac.jp (T.O.); s-takise@dokkyomed.ac.jp (S.T.); j-fujita@dokkyomed.ac.jp (J.F.); shmorita@dokkyomed.ac.jp (S.M.); kojima-k@dokkyomed.ac.jp (K.K.)

**Keywords:** esophagogastric junction cancer, surgical approach, lymph node dissection, minimally invasive surgery, perioperative treatment, immunotherapy, treatment standardization, regional disparities

## Abstract

The number of esophagogastric junction cancers (EGJCs) has been rising globally, yet its optimal treatment remains controversial due to its complex anatomical location. This review outlines the current evidence on surgical strategies, lymph node dissection, and perioperative therapies, with a focus on differences between Eastern and Western clinical practices. By integrating data from major global trials, this article aims to clarify regional trends and guide future standardized approaches to EGJC.

## 1. Introductions

The incidence of esophagogastric junction cancer (EGJC) is rising globally, with notable increases in both Western and Eastern countries [1]. Traditionally, EGJC has been managed based on treatment strategies for either esophageal or gastric cancer. However, due to its unique anatomical location, complex lymphatic drainage, and diverse histological features, a distinct approach to EGJC is warranted.

Regional differences in classification systems further complicate its management. The Siewert classification is commonly used in Western countries, while the Nishi classification is predominant in Japan [2,3]. Moreover, treatment paradigms vary significantly between regions. Eastern countries such as Japan and Korea have relied on surgery followed by adjuvant chemotherapy, supported by robust outcomes from D2 lymphadenectomy [4,5,6]. In contrast, Western countries like Germany, France, and the United States have adopted perioperative or neoadjuvant approaches, including chemoradiotherapy, based on trials such as FLOT4 and CROSS [7,8].

This review aims to summarize and critically appraise the current evidence regarding the treatment of EGJC, focusing on three major components: surgical approach, lymph node dissection, and perioperative treatment. Special emphasis is placed on regional differences, ongoing clinical trials, and future directions for harmonizing strategies across countries.

Furthermore, these regional differences may stem not only from clinical evidence but also from variations in healthcare infrastructure, cultural norms, and historical treatment paradigms, which complicate efforts toward global standardization.

## 2. Surgical Treatment

### 2.1. Surgical Approach

Several randomized controlled trials (RCTs) have investigated the optimal surgical approach for EGJC, focusing on balancing oncological efficacy and minimizing postoperative complications. Table 1 summarizes key RCTs comparing different approaches.

Early studies such as those by Hulscher et al. and Sasako et al. demonstrated that while the transthoracic or thoracoabdominal approaches might offer a theoretical advantage in nodal clearance, they are often associated with higher pulmonary complication rates and do not significantly improve overall survival (OS) in Siewert type II tumors [9,10]. More recent trials, including the TIME and MIRO studies, have suggested that minimally invasive surgery (MIS) can reduce postoperative morbidity without compromising oncological outcomes [11,12]. Notably, the RAMIE trial indicated significantly lower cardiopulmonary complications with robotic-assisted esophagectomy [13].

While MIS approaches offer technical advantages and improved recovery profiles, their success heavily depends on surgeon experience and institutional expertise. Real-world evidence and ongoing trials (e.g., ROBOT-2) will further clarify their role in routine EGJC management [14,15].

### 2.2. Optimal Extent of Lymph Node Dissection

Due to the complex lymphatic drainage of the esophagogastric junction, the optimal extent of lymphadenectomy remains controversial. Table 2 highlights major studies evaluating nodal spread patterns.

Most studies agree on the significance of lower mediastinal and upper perigastric node involvement, particularly No. 110 [16,17,18,19]. Kurokawa et al.’s 2021 prospective study introduced a stratified approach to lymphadenectomy based on the length of esophageal invasion, which is now incorporated into Japanese guidelines [20]. Yoshikawa et al. proposed that limited dissection may suffice in short tumors with minimal esophageal invasion [21].

These findings underscore the importance of individualized dissection strategies tailored to tumor length and location, pending further validation through survival outcomes.

## 3. Pre- and Postoperative Treatment

A summary of representative clinical trials on pre- and postoperative chemotherapy, chemoradiotherapy, and immunotherapy in EGJC is provided in Table 3.

### 3.1. Pre- and Postoperative Chemotherapy

In Eastern Asia, upfront surgery followed by adjuvant chemotherapy has traditionally been favored for EGJC [4,5,6]. However, recent trials from both Eastern and Western regions have evaluated the efficacy of chemotherapy administered before and/or after surgery.

The FLOT4 trial in Germany demonstrated improved overall survival using a triplet regimen before and after surgery [7]. Similarly, the RESOLVE trial in China and the PRODIGY trial in Korea showed that incorporating systemic therapy before surgery improved disease-free or progression-free survival, though overall survival benefits were inconsistent [22,23]. Recently, an RCT conducted by the Japan Clinical Oncology Group was initiated to compare preoperative chemotherapy of docetaxel, oxaliplatin, and S-1 (DOS regimen) to that of fluorouracil, oxaliplatin, and docetaxel (FLOT regimen) [24]. Both groups are followed by surgery and adjuvant chemotherapy with S1 ± docetaxel. The results of this RCT are expected to further inform the treatment strategy for EGJC.

These findings suggest that chemotherapy delivered in both pre- and postoperative settings may improve surgical outcomes, but its survival benefit may depend on regimen selection, tumor biology, and regional practices.

### 3.2. Pre- and Postoperative Chemoradiotherapy

Chemoradiotherapy prior to surgery has been investigated as an alternative approach to improve locoregional control and resectability. The CROSS trial established its value in Western countries, demonstrating improved R0 resection and survival outcomes. [8] However, the results remain inconsistent. The POET trial in Germany indicated improved pathological response and a trend toward longer survival with chemoradiotherapy versus chemotherapy alone. In contrast, the NeoRes trial in Scandinavia reported no survival advantage, despite improved R0 resection and nodal downstaging. [25,26] These conflicting results may be attributed to differences in chemotherapy regimens, radiation dosing, and patient selection. While chemoradiotherapy may enhance local control and pathological response, its survival benefit remains context-dependent.

### 3.3. Pre- and Postoperative Immunotherapy

Immunotherapy has emerged as an important addition to the multimodal treatment of EGJC. The CheckMate 577 trial demonstrated a significant improvement in disease-free survival with adjuvant nivolumab following neoadjuvant chemoradiotherapy and surgery, establishing its role in the Western treatment paradigm [27]. In contrast, the ATTRACTION-5 trial from Asia failed to show survival benefits for adjuvant nivolumab combined with chemotherapy, suggesting that immunotherapy may be more effective in the presence of residual disease post-chemoradiotherapy [28].

Recent global trials such as KEYNOTE-585 and MATTERHORN have investigated the addition of PD-1/PD-L1 inhibitors to neoadjuvant and adjuvant chemotherapy. These trials reported increased pathological complete response rates, particularly in patients with PD-L1–positive tumors, though survival benefits are still under evaluation [29,30]. Notably, biomarker-driven approaches, including assessments of PD-L1 expression and microsatellite instability (MSI), have become critical in selecting patients who are likely to benefit from immunotherapy [31].

In metastatic or unresectable EGJC, immune checkpoint inhibitors have become a standard of care. Trials such as KEYNOTE-590 and CheckMate 649 have shown that combining immunotherapy with chemotherapy improves overall survival, particularly in PD-L1-positive populations [32,33]. Furthermore, HER2-positive and PD-L1-positive cases benefit from the addition of pembrolizumab to trastuzumab and chemotherapy, as demonstrated in KEYNOTE-811 [34].

Overall, checkpoint inhibitors have expanded treatment options for EGJC, especially when guided by predictive biomarkers. As evidence accumulates, immunotherapy is likely to become a central component of both curative and palliative treatment strategies.

## 4. Conclusions

The treatment strategy for EGJC continues to evolve in response to accumulating clinical evidence and regional practice differences. This review has highlighted that surgical approach, extent of lymphadenectomy, and pre- and postoperative therapies must be tailored based on tumor characteristics and geographic context.

Minimally invasive techniques and stratified lymphadenectomy protocols have improved safety without compromising curability. Emerging evidence supports the use of pre- and postoperative therapy, with regional variation still playing a substantial role. East Asian countries are increasingly integrating preoperative chemotherapy, while Western nations continue to lead in chemoradiotherapy and immunotherapy adoption. Future efforts should aim to unify global treatment strategies through collaborative research, better understanding of tumor biology, and equitable access to multimodal therapies. EGJC, as a junctional cancer, demands a unified yet flexible approach that bridges East–West paradigms for optimal outcomes worldwide.

Clinicians may benefit from a stepwise approach that accounts for tumor location, histological subtype, nodal involvement, and patient fitness in selecting optimal treatment strategies. For researchers, international, multicenter trials that incorporate stratification by region and treatment environment are strongly encouraged to overcome disparities and support global consensus-building. Taken together, understanding the sociocultural, economic, and institutional drivers behind regional strategies is essential not only for interpreting existing data but also for designing future clinical trials that aim for broader international applicability. To further clarify the regional heterogeneity in EGJC management, Figure 1 provides a comparative summary of classification systems, surgical approaches, lymphadenectomy strategies, and perioperative treatments between Eastern and Western countries. This visualization highlights both commonalities and key distinctions informed by major clinical trials. Such comprehensive comparisons can support clinicians and policymakers in identifying region-adapted but evidence-based treatment pathways. Ultimately, advancing EGJC care will require not only scientific innovation but also sustained efforts to integrate evidence into practice across diverse healthcare settings.

## Figures and Tables

**Figure 1 cancers-17-01961-f001:**
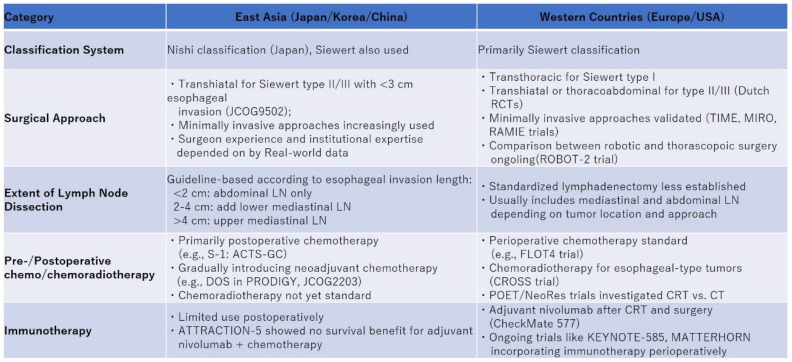
Comparison of treatment strategies for esophagogastric junction cancer (EGJC) between Eastern and Western countries. This figure summarizes the differences and similarities in the management of EGJC between Eastern and Western clinical practices. Key elements compared include classification systems (Siewert vs. Nishi), preferred surgical approaches (transhiatal vs. transthoracic), extent of lymph node dissection (based on esophageal invasion length in the East), and perioperative treatments. While the West favors neoadjuvant chemoradiotherapy and immune checkpoint inhibitors based on trials like CROSS and CheckMate 577, the East has adopted preoperative or postoperative chemotherapy through trials such as RESOLVE and PRODIGY.

**Table 1 cancers-17-01961-t001:** Randomized controlled trials regarding surgical approach for EGJC.

Author	Hulscher	Sasako	Biere	Mariette	van der Sluise
Country	Netherlands	Japan	Netherlands	France	Netherlands
Year	2007	2006	2017	2019	2019
Number of patients	220	167	115	207	112
Surgical approach ①	Transhiatal	Transhiatal	Thoracoscopy/Laparoscopy	Thoracotomy/Laparoscopy	RAMIE/Laparoscopy
Surgical approach ②	Right transthoracic	Left thoracoabdominal	Thoracotomy/Laparotomy	Thoracotomy/Laparotomy	Thoracotomy/Laparoscopy
Tumor location					
Upper/middle esophagus	0	0	45.2%	30.9%	12.8%
Lower esophagus/EGJC	100%	95.8%	54.8%	69.1%	87.2%
Siewert type I	43.9%	0	NA	NA	NA
Siewert type II	56.1%	57.6%	NA	NA	NA
Siewert type III	0	38.2%	NA	NA	NA
Stomach	0	4.2%	0	0	0
Histological type					
AC	96.1%	100%	61.7%	59.4%	77.1%
SCC	3.9%	0	37.4%	40.6%	22.9%
Other	0	0	0.90%	0	0
Neoadjuvant treatment					
Chemoradiotherapy	0	0	92.2%	31.9%	79.5%
Chemotherapy	0	0	7.8%	41.5%	8.9%
None	100%	100%	0	26.6%	11.6%
R0 resection	71.6% vs. 71.8% (*p* = 0.28)	92.7% vs. 88.2%	91.5% vs. 83.9%	95.1% vs 98.1%	92.5% vs. 96.4% (*p* = 0.35)
Operation time (min)	210 vs. 360 (*p* < 0.001)	305 vs. 338	329 vs. 299 min (*p* = 0.002)	327 vs 330 min	349 vs. 296 min (*p* < 0.001)
Blood loss (mL)	1000 vs. 1900 (*p* < 0.001)	673 vs. 655	200 vs. 475 mL (*p* < 0.001)	NA	400 vs. 568 mL (*p* < 0.001)
Open conversion rate	NA	NA	13.6%	2.9%	5.4%
Complication	NA	34.1% vs. 49.4% (*p* = 0.06)	NA	35.9% vs 64.4%	59.2 vs. 80.0% (*p* = 0.02)
Anastomotic leakage	14.1% vs. 15.8% (*p* = 0.85)	6.1% vs. 8.2% (*p* = 0.77)	11.9% vs. 7.1% (*p* = 0.390)	10.8% vs. 6.8%	24.0% vs. 20.0% (*p* = 0.57)
Pulmonary complications	27.3% vs. 57.0% (*p* < 0.001)	3.7% vs. 12.9% (*p* = 0.05)	11.9% vs. 33.9% (*p* = 0.005)	17.6% vs. 30.1%	31.5% vs. 58.2% (*p* = 0.005)
Cardiac	16.0% vs. 26.3% (*p* = 0.10)	NA	NA	11.8% vs. 13.4%	22.2% vs. 47.3% (*p* = 0.006)
Vocal cord paralysis	13.2% vs. 21.1% (*p* = 0.15)	NA	1.7% vs. 14.3% (*p* = 0.012)	NA	9.3% vs. 10.9% (*p* = 0.78)
Chylous leakage	1.9% vs. 9.6% (*p* = 0.02)	NA	NA	4.9% vs. 6.8%	31.5% vs. 21.8% (*p* = 0.69)
Pancreatic fistula	NA	12.1% vs. 16.5% (*p* = 0.51)	NA	NA	NA
Abdominal Abscess	NA	8.5% vs. 14.1% (*p* = 0.33)	NA	NA	NA
Pyothorax	NA	1.2% vs. 4.7% (*p* = 0.37)	NA	NA	NA
Mediastinitis	NA	0 vs. 4.7% (*p* = 0.12)	NA	NA	NA
Reoperation	NA	NA	13.6% vs. 10.7% (*p* = 0.641)	NA	24.0% vs. 32.7% (*p* = 0.32)
Mortality	1.9% vs. 4.4% (*p* = 0.45)	0% vs. 5.9% (*p* = 0.25)	1.7% vs. 0 (*p* = 0.590)	1.0% vs. 1.9%	1.8% vs. 0 (*p* = 0.62)
Survival	5-year OS Siewert type I 37% vs. 51% (*p* = 0.33) Siewert type II 31% vs. 27% (*p* = 0.81)	5-year OS Siewert type II 50% vs. 42% (*p* = 0.496) Siewert type III 59% vs. 36% (*p* = 0.102)	3-year OS 42.9% vs. 41.2% (*p* = 0.633) 3-year DFS 37.3% vs. 42.9% (*p* = 0.602)	5-year OS 60% vs. 40% (HR 0.67, 95%CI 0.44–1.01) 5-year DFS 53% vs. 43% (HR 0.76, 95%CI 0.52–1.11)	Median DFS 26 vs. 28 months (*p* = 0.983)

EGJC: esophagogastric junction cancer; RAMIE: robot-assisted minimally invasive esophagectomy; AC: adenocarcinoma; SCC: squamous cell carcinoma; NA: not available; OS: overall survival; DFS; disease-free survival.

**Table 2 cancers-17-01961-t002:** Summary of articles regarding lymph node metastasis of patients with EGJC.

Author	Siewert	Pedrazzani	Kurokawa	Yoshikawa	Yamashita	Kurokawa
Country	Germany	Italy	Japan	Japan	Japan	Japan
Year	2000	2007	2015	2016	2017	2021
Study design	Retrospective	Retrospective	Retrospective	Retrospective	Retrospective	Prospective
Number of patients	271	62	315	381	2807	363
Definition of EGJC	Siewert type II	Siewert type II	Siewert type II	Siewert type II	Nishi classification	Nishi classification
Eligibility	pT1-4	pT2-4	pT2-4	pT1-4	pT1-4 tumor size ≤ 4 cm	cT2-4
Histological type						
SCC	0	0	0	0	13.2	8.5
AC	100	100	100	100	84.9	91.5
Other	0	0	0	0	1.9	0
Tumor size, mm *	NA	NA	55 (8–100)	50 (10–180)	25 (16–39)	46 (10–150)
Preoperative treatment, %	22.6	0	14.0	10.8	0	33.3
pT status						
T0	0	0	0	0	0	4.4
T1	14.0	0	0	20.7	56.6	13.2
T2	57.2	51.6	18.1	14.7	19.2	16.5
T3	20.3	46.8	45.1	36.0	24.1	48.2
T4	8.5	1.6	36.8	28.6	(T3 and T4)	16.3
pN status						
N0	31.4	29.0	23.8	35.7	69.5	30.6
N1	29.5	71.0 (N positive)	21.6	20.7	16.7	25.1
N2	22.5		27.6	22.6	9.0	20.9
N3	16.6		27.0	21.0	4.8	22.0
pM status						
M0	83.8	100	100	93.2	100	96.1
M1	16.2	0	0	6.8	0	3.9
Esophagectomy						
Total/subtotal	NA	NA	7.0	7.1	NA	35.5
Lower/abdominal	predominated	NA	93.0	92.9	NA	64.5
Gastrectomy						
Total	NA	NA	77.1	69.3	NA	49.0
Proximal/upper	NA	NA	22.9	30.7	NA	51.0
Metastatic lymph nodes, %						
Upper mediastinal nodes	NA	NA	3.8	NA	0.0–5.1	6.1
No. 105	NA	NA	NA	NA	0.0–1.1	1
No. 106recL	NA	NA	NA	NA	NA	1
No. 106recR	NA	NA	NA	NA	0.0–5.1	5.1
No. 106tb	NA	NA	NA	NA	0	NA
Middle mediastinal nodes	NA	NA	7.0	NA	0.0–4.0	7.1
No. 107	NA	1.6	NA	NA	0.0–1.7	3.1
No. 108	NA	<5.0	NA	NA	0.8–4.0	5.1
No. 109	NA	NA	NA	NA	0.0–2.8	NA
No. 109L	NA	NA	NA	NA	NA	3.1
No. 109R	NA	NA	NA	NA	NA	2.0
Lower mediastinal nodes	15.6	NA	11.4	NA	0.3–11.9	13.3
No. 110	NA	12.9	NA	NA	0.5–11.9	9.3
No. 111	NA	5.0–10.0	NA	NA	0.3–3.4	3.4
No. 112	NA	5.0–10.0	NA	NA	0.0–2.3	2.0
Abdominal nodes	NA	NA	NA	NA	NA	NA
No. 1	56.9	50	NA	39.8	4.0–34.6	35.2
No. 2	67.8	30–35	NA	30.8	1.6–16.5	27.1
No. 3	67.8	50–55	NA	41.5	3.9–39.5	38
No. 4	16.1	NA	NA	NA	NA	NA
No. 4sa	NA	0–5.0	NA	4.3	0.1–0.3	4.2
No. 4sb	NA	NA	NA	2.7	0.0–1.3	0.8
No. 4d	NA	5.0–10.0	NA	2.9	0.0–0.8	2.2
No. 5	1.6	0–5.0	NA	1.7	0.0–0.5	1.1
No. 6	NA	0–5.0	NA	0.8	0.0–0.9	1.7
No. 7	15.1	30.6	NA	26.7	1.1–17.7	23.5
No. 8a	NA	15–20	NA	4.9	0.2–3.8	7.1
No. 9	7	15–20	NA	11.7	0.3–6.8	12.4
No. 10	NA	0–5.0	NA	9.5	0.1–0.9	NA
No. 11	4.8	5.0–10.0	NA	NA	NA	NA
No. 11p	NA	NA	NA	17.2	0.3–4.5	13.6
No. 11d	NA	NA	NA	6.3	0.0–2.1	4.3
No. 12	4.8	0	NA	1.4	NA	NA
No. 16a1	NA	0–5.0	NA	NA	0.0–0.3	NA
No. 16a2	NA	NA	NA	14.4	0.0–0.6	4.7
No. 19	NA	NA	NA	6.3	0.0–0.8	5.4
No. 20	NA	NA	NA	0.0	0.0–0.8	4.8

EGJC: esophagogastric junction cancer; SCC: squamous cell carcinoma; AC: adenocarcinoma; NA: not available. * Values are shown as median and range.

**Table 3 cancers-17-01961-t003:** Clinical trials regarding pre- and postoperative treatment for EGJC.

Types of Treatment	Chemotherapy			Chemoradiotherapy			Immunotherapy
Study name	FLOT4	PRODIGY	RESOLVE	CROSS	POET	NeoRes	Checkmate577
Country	Germany	South Korea	China	Netherlands	Germany	Norway, Sweden	29 countries
Phase	II/III	III	III	III	III	II	III
Eligibility	cT2–4 or cN(+)	cT2–3 cN(+) or cT4	cT4a cN(+) or cT4b	cT1N1M0 or cT2–3N0–1M0	cT3–4NXM0	cT1N(+)M0 or cT2–3NXM0	ypStage II-III after pre-CRT plus surgery
Experimental arm	Pre- and Post-FLOT	Pre-DOS and Post S-1	Pre- and post-SOX	Pre-(CBDCA+PTX+RT)	Pre-(FP+RT)	Pre-(PLF+RT)	Post-nivolumab
Control arm	Pre- and Post-ECF/ECX	Post S-1	Post CAPEOX	Surgery alone	Pre-FP	Pre-PLF	None
Number of patients	716	484	1022	368	119	181	794
Histological type							
AC	100%	100%	NA	75%	100%	72%	71%
SCC	0	0	NA	25%	0	28%	29%
Unknown	0	0	NA	0	0	0	0.10%
Tumor location							
Esophagus	0	0	0	73%	0	83%	58%
EGJ	56%	6%	36%	24%	100%	17%	42%
Stomach	44%	94%	64%	0	0	0	0
Unknown	0	0	0	3%	0	0	0
cT status							
T0	0	0	3% (T1–T3)	0	0	0	6%
T1	1%	0		1%	0	1%	39%
T2	15%	5%		17%	0	34%	0
T3	73%	24%		81%	92%	65%	55%
T4	9%	71%	97%	0.3%	8%	0	0
Unknown	3%	0	NA	1%	0	0	0.4%
cN status							
N(−)	21%	2%	NA	32%	NA	37%	42%
N(+)	79%	98%	NA	64%	NA	63%	58%
Other	0	0	NA	0	NA	0	0.1%
Survival	Median OS 50 vs. 35 months 3-year OS rate 57% vs. 48% 5-year OS rate 45% vs. 36% (HR 0.77, 95% CI 0.63–0.94)	3-year PFS 66% vs. 60% 5-year PFS 60% vs. 56% (HR 0.70, 95% CI 0.52–0.95, *p* = 0.02)	3-year OS 59.4% vs. 51.1% (HR 0.77, 95% CI 0.61–0.97)	Median OS 49 vs. 24 months (*p* = 0.003, HR 0.657, 95% CI 0.495–0.871) 3-year OS 58% vs. 44% 5-year OS 47% vs. 33% (HR 0.68, 95% CI 0.53–0.88) 3-year PFS 51% vs. 35% 5-year PFS rate 44% vs. 27%	Median OS 33 vs. 21 months 3-year OS 47% vs. 28% (HR 0.67, 95% CI 0.41–1.07)	3-year OS 47% vs. 49% (HR 1.09, 95% CI 0.73–1.64) 3-year PFS 44% vs. 44%	Median DFS 22.4 vs. 11.0 months (HR 0.69 96.4% CI 0.56–0.86, *p* < 0.001)

AC: adenocarcinoma; SCC squamous cell carcinoma; NA: not available; EGJ: esophagogastric junction; OS overall survival; HR hazard ratio; CI confidence interval; PFS progression free survival; DFS disease free survival.

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
