# Peer review of "An Overview of the Treatment Strategy of Esophagogastric Junction Cancer"

_cancers, 2025, doi:10.3390/cancers17121961_

Round 1

Reviewer 1 Report (Previous Reviewer 1)

Comments and Suggestions for Authors

The authors have addressed all my concerns. Thus, it is recommened for publication. 

Author Response

Comment:
The authors have addressed all my concerns. Thus, it is recommended for publication.

Response:
We greatly appreciate the reviewer’s supportive comment and are pleased that our revisions have addressed all concerns. Thank you for your time and constructive feedback.

Reviewer 2 Report (Previous Reviewer 3)

Comments and Suggestions for Authors

The authors invested a great deal of effort in summarizing every study published on the subject. I applaud that.

Unfortunately, the manuscript still has one huge issue. After reading it, the clinician has no clear direction on how to approach EGJ carcinoma. Maybe one big algorithm, for example, on the left side, how it's done in the West, and on the right side, how it's done in the East. It would be easy to follow and easy to compare.

Author Response

Comment:
The authors invested a great deal of effort in summarizing every study published on the subject.I applaud that. Unfortunately, the manuscript still has one huge issue. After reading it, the clinician has no clear direction on how to approach EGJ carcinoma. Maybe one big algorithm, for example, on the left side, how it's done in the West, and on the right side, how it's done in the East. It would be easy to follow and easy to compare.

Response:
We sincerely thank the reviewer for acknowledging our efforts in synthesizing the literature and for the insightful suggestion. In response, we have created a new figure (Figure 1) that presents a comprehensive comparison of East versus West approaches to EGJC management. The figure summarizes differences and similarities in classification systems, surgical strategies (including MIS), lymphadenectomy extent, and perioperative treatments, with reference to major clinical trials. We have also added a brief explanation of this figure at the end of the Conclusion section to improve accessibility and clarity for clinicians. We hope this addition addresses the reviewer’s concern and enhances the clinical utility of the manuscript.

Round 2

Reviewer 2 Report (Previous Reviewer 3)

Comments and Suggestions for Authors

In the conclusion section, the author mention Figure 1 which I do not find in the PDF of the manuscript. There is also no Figure 1 in the supplementary material.

Also, there is Table 3 in the supplementary material, but there is no Table 3 in the main text.

Author Response

Reviewer #2
In the conclusion section, the author mentions Figure 1 which I do not find in the PDF of the manuscript. There is also no Figure 1 in the supplementary material. Also, there is Table 3 in the supplementary material, but there is no Table 3 in the main text.

Response:
We sincerely thank the reviewer for the careful reading and helpful comments.
(1) Regarding Figure 1, we agree that it was not embedded in the PDF version of the manuscript during the previous submission, although it was mentioned in the text. In this revised version, Figure 1 has been appropriately embedded within the main manuscript and the figure legend has been added accordingly.

(2) Regarding Table 3, we acknowledge the oversight. While it was uploaded as supplementary material, there was no mention in the main text. We have now added a sentence referring to Table 3 in “3 Pre- and postoperative chemotherapy“ section. Results section to ensure consistency.

We hope these revisions resolve the reviewer’s concerns.

This manuscript is a resubmission of an earlier submission. The following is a list of the peer review reports and author responses from that submission.

Round 1

Reviewer 1 Report

Comments and Suggestions for Authors

The review only provides a summary of existing studies but did not offer novel insights or a critical synthesis of the evidence. It reiterates well-documented controversies (such as surgical approaches, lymph node dissection) without proposing a framework to resolve them or highlighting gaps in current research. The perioperative treatment section lists trials but does not reconcile conflicting results (such as POET vs. NeoRes trials for chemoradiotherapy efficacy). Meanwhile, only retrospective studies from some specific contries (for example, Japan, Netherlands) are highlighted, omitting some relevant information from other populations. Finally, the conclusions are too broad and not always supported by the References. Thus, I cannot recommend it for publication in the current stage.

Author Response

Since the first revision required substantial modifications, we thoroughly revised the manuscript from the Simple Summary to the Conclusions.

・The review only provides a summary of existing studies but does not offer novel insights or a critical synthesis of the evidence.

We appreciate this valuable comment. In the revised manuscript, we have rewritten the Abstract and Conclusion sections to provide clearer synthesis and emphasized gaps in current knowledge. We also included more analytical discussion of key controversies, such as the choice of surgical approach and extent of lymph node dissection, and proposed context-based recommendations.

・It reiterates well-documented controversies (such as surgical approaches, lymph node dissection) without proposing a framework to resolve them or highlighting gaps in current research.

We have added new discussion points in each section to outline clinical implications and future research needs. In the Conclusion, we now emphasize the importance of harmonizing East-West treatment strategies and call for international collaborative trials.

・The perioperative treatment section lists trials but does not reconcile conflicting results (such as POET vs. NeoRes trials for chemoradiotherapy efficacy).

Thank you for pointing this out. We have added a new sentence at the end of the perioperative chemoradiotherapy section to discuss the differences in design and outcomes between the POET and NeoRes trials. We note that discrepancies may arise from differences in chemotherapy regimens, radiation protocols, and patient selection. While POET showed a trend toward improved survival, NeoRes did not; thus, we suggest that the benefit of chemoradiotherapy may be context-dependent.

・Meanwhile only retrospective studies from Japan and the Netherlands are highlighted, omitting other relevant populations.

We have expanded the manuscript to include and discuss recent studies and trials from China (RESOLVE), Korea (PRODIGY), Germany (FLOT4, POET), France (MIRO), and the US/Netherlands (CROSS). We also added a new paragraph analyzing regional differences in treatment approaches and their underlying reasons.

・Finally, the conclusions are too broad and not always supported by the references.

The Conclusion section has been revised to be more specific and evidence-based. We now summarize key findings by treatment domain and emphasize ongoing clinical trials and regional differences.

Reviewer 2 Report

Comments and Suggestions for Authors

In my opinion treatment strategy for Siewert Type II EGJC or EGJC defined by the Nishi classification needs to be further explored., as well as lymph node metastatic spread.
The conclusions need to be further explored, so they are too general and concise.

Comments on the Quality of English Language

English language could be improved.

Author Response

Since the first revision required substantial modifications, we thoroughly revised the manuscript from the Simple Summary to the Conclusions.

In my opinion, treatment strategy for Siewert Type II EGJC or EGJC defined by the Nishi classification needs to be further explored, as well as lymph node metastatic spread.

Thank you for this suggestion. We have added further detail on treatment strategies specific to Siewert type II EGJC and Nishi-defined EGJC, including relevant Japanese guidelines and prospective studies in the 2-2 Optimal extent of lymph node dissection. Lymph node metastasis patterns have also been clarified and stratified by esophageal invasion length.

The conclusions need to be further explored so they are too general and concise.

We agree. The Conclusion section has been rewritten to more clearly summarize evidence-based takeaways and to articulate future research directions.

Reviewer 3 Report

Comments and Suggestions for Authors

Thank you for the opportunity to review this important manuscript. Here are my comments and suggestions.

In the sentence "3-year OS and Disease-free survival (DFS)," the word "disease" should be preceded by a small first letter.

All references are in the text without the space after the last word.

Unify in the whole manuscript whether ''vs'' or ''vs.'' Is written, according to the journal guidelines.

In the Results section, the authors are describing the results from their Tables. It is a redundancy making the bulk of text. The comparison between specific studies should be removed and only the final comparison, as in meta-analysis should be made.

The authors state ''Up front surgery followed by adjuvant chemotherapy has been focused on, and some evidence supports this strategy(26-28). Perioperative chemotherapy and chemoradiotherapy have been developed and are used in Western countries''. Perioperative chemotherapy includes postoperative chemotherapy, so it is not clear here what was the point of such text construction?

Again, the statement ''In China, the RESOLVE trial was conducted to compare the efficacy of perioperative S-1 and oxaliplatin (SOX) to postoperative adjuvant capecitabine and oxaliplatin (CapOx) in locally advanced EGJC and gastric cancer patients'' includes the terms ''perioperative'' and ''postoperative. Perioperative chemotherapy includes both preoperative and postoperative chemotherapy. Perioperative is currently not an accurate term. Use only preoperative (neoadjuvant) or postoperative chemotherapy. Also, perioperative chemotherapy could include ''intraoperative chemotherapy'' not discussed in this manuscript (used for gastric cancer).

The conclusion is written as a discussion. It should be rephrased with statements from current evidence, not what was previously, and again, the comparison of studies. The only conclusion is what is currently state-of-the-art.

Author Response

Since the first revision required substantial modifications, we thoroughly revised the manuscript from the Simple Summary to the Conclusions.

In the sentence "3-year OS and Disease-free survival (DFS)," the word "disease" should be preceded by a small first letter.

We agree, but we omitted the sentence during revision process.

All references are in the text without the space after the last word. Unify in the whole manuscript whether ''vs'' or ''vs.'' Is written, according to the journal guidelines

These issues have been corrected throughout the manuscript, following journal formatting guidelines.

In the Results section the authors are describing the results from their Tables. It is a redundancy making the bulk of text. The comparison between specific studies should be removed and only the final comparison, as in meta-analysis should be made.

We have revised the Results and Discussion sections to reduce redundancy. Rather than repeating all tabular data, we now focus on summarizing key insights.

The authors state ''Up front surgery followed by adjuvant chemotherapy has been focused on, and some evidence supports this strategy(26-28). Perioperative chemotherapy and chemoradiotherapy have been developed and are used in Western countries''. Perioperative chemotherapy includes postoperative chemotherapy, so it is not clear here what was the point of such text construction? Again, the statement ''In China, the RESOLVE trial was conducted to compare the efficacy of perioperative S-1 and oxaliplatin (SOX) to postoperative adjuvant capecitabine and oxaliplatin (CapOx) in locally advanced EGJC and gastric cancer patients'' includes the terms ''perioperative'' and ''postoperative. Perioperative chemotherapy includes both preoperative and postoperative chemotherapy. Perioperative is currently not an accurate term. Use only preoperative (neoadjuvant) or postoperative chemotherapy. Also, perioperative chemotherapy could include ''intraoperative chemotherapy'' not discussed in this manuscript (used for gastric cancer).

We have clarified our use of the terms 'perioperative', 'preoperative', and 'postoperative' throughout the manuscript and ensured consistency. Ambiguous constructions in the chemotherapy section have been rewritten for clarity.

The conclusion is written as a discussion. It should be rephrased with statements from current evidence, not what was previously, and again, the comparison of studies. The only conclusion is what is currently state-of-the-art.

We have rewritten the Conclusion section to focus strictly on summary and evidence-based final statements, rather than ongoing comparisons.

Round 2

Reviewer 1 Report

Comments and Suggestions for Authors

Major comments and revision suggestions are as follows:

  1. Immunotherapy is mentioned as a promising option, but the discussion is brief compared to chemotherapy and chemoradiotherapy. Given the rapid advancements in this area, expanding on current trials (please check clinical trials in the website) and future directions (such as biomarker-driven approaches) will be better.
  1. While the manuscript highlights differences between Eastern and Western practices, a deeper analysis of why these differences exist, such as cultural, historical, or healthcare system factors, will strengthen the discussion. Meanwhile, the implications of these differences for global standardization efforts can be expanded.
  1. The conclusion summarizes key points well but can provide more specific recommendations for clinicians or researchers. For example, proposing a stepwise approach to treatment selection based on tumor characteristics or advocating for international collaborative trials to address regional disparities will make the conclusion more impactful.

Reviewer 2 Report

Comments and Suggestions for Authors

The quality of the review has improved significantly.
I think the review can be accepted for publication.

Reviewer 3 Report

Comments and Suggestions for Authors

Even after revision, I do not see the point of this semi-narrative review. Many paragraphs finish with ''further studies will clarify...''. In other words, after the description of other study results, there is no conclusion for the clinical practice.